# Smartphone-Based Color Evaluation of Passive Samplers for Gases: A Review

**Kanokwan Kiwfo** [1,2], **Kate Grudpan** [1,3] , **Andreas Held** [2] **and Wolfgang Frenzel** [2,*]

1 Center of Excellence for Innovation in Analytical Science and Technology for Biodiversity-Based Economic and Society (I-ANALYS-T_B.BES-CMU), Chiang Mai University, Chiang Mai 50200, Thailand; k.kanokwan11@gmail.com (K.K.); kgrudpan@gmail.com (K.G.)
2 Environmental Chemistry and Air Research, Institute of Environmental Technology, Technische Universität Berlin, Str. d. 17. Juni 135, D-10623 Berlin, Germany; held@tu-berlin.de
3 Department of Chemistry, Faculty of Science, Chiang Mai University, Chiang Mai 50200, Thailand
* Correspondence: wolfgang.frenzel@tu-berlin.de

**Abstract:** The application of smartphone-based color evaluation of passive sampling devices for gases has only been sparsely reported. The present review aims to compile available publications with respect to the configuration of the passive samplers, conditions of smartphone photographing, analytical procedures for color detection and quantification (including calibration processes), and their application to different target gases. The performance of the methods—whenever available—is presented regarding the analytical specifications selectivity, sensitivity, and limit of detection in comparison with other color evaluation methods of passive samplers. Practical aspects like requirements of instrumentation and ease of use will be outlined in view of the potential employment in education and citizen science projects. In one section of the review, the inconsistent terminology of passive and diffusive sampling is discussed in order to clarify the distinction of information obtained from the uptake of the passive samplers between gas-phase concentration and the accumulated deposition flux of gaseous analytes. Colorimetric gas sensors are included in the review when applied in passive sampling configurations and evaluation is performed with smartphone-based color evaluation. Differences in the analytical procedures employed after the passive sampling step and prior to the detection of the colored compounds are also presented.

**Keywords:** passive gas sampling; diffusive sampling; ambient air measurements; immission rate; smartphone-based color evaluation; RGB color space; colorimetric gas sensors

## 1. Introduction

Passive sampling of gases is a frequently applied method for the determination of a variety of gases. The methodology and applications are the subject matter of many reviews [1–7], innumerous original papers (a recent search in the ScifinderSearch database revealed about 3600 hits for the combination of the terms "Passive/Diffusive sampling", "Gas analysis", and "Air pollution" in the past four decades) as well as reports by governmental institutions [8–10] and non-governmental organizations [11–13]. The attractive features of passive gas sampling devices for gas analyses are that they do not need external power, operate silently, are small in size, are lightweight, and are cheap to construct. They are also readily deployable at almost any place by nonspecialists. Simultaneous gas sampling in an affordable way at many sites is hence possible, providing high spatial resolution. One limitation of many passive sampling methods can be seen in the need for considerably long sampling times (typically days to months), providing only time-weighted integrated values. Nonetheless, useful and frequent applications can be found in large-scale mapping, long-term personal exposure estimates, and identification of air pollution hot spots [14–19]. However, many examples of passive sampling also exist for short-term measurements of time-weighted exposure to gaseous contaminants applied to personal monitoring [20–22].

In this context, the term passive dosimeters (rather than passive samplers) is often used to indicate that a certain dose of contaminants is inhaled when air is aspirated.

Historically, passive gas sampling can be traced back to 1853 when the Swiss chemist Schönbein used filter papers impregnated with potassium iodide to measure ground-level ozone (cited in [23]). At the beginning of the 20th century, $SO_2$ was detected using the darkening of a freely hanging lead acetate paper [24]. Visual inspection provided qualitative and, to a certain degree (using a color scale for comparison), even semi-quantitative information. Further early examples of passive gas sampling are the Liesegang–Glockenverfahren [25] and the $SO_2$ cylinder (also called wick and candle) ([26] and references therein). In these methods, $SO_2$ is trapped on solid lead peroxide, and the sulfate formed is subsequently determined by means of gravimetry, titrimetry, or turbidimetry. All these examples rely on sampling of the analyte gases by molecular diffusion to the absorbing surfaces. Hence, the term passive sampling (no pump required for active air transport) or synonymously diffusive sampling is appropriate and has been established for distinction from active sampling since. The diffusional transport is governed by the concentration gradient between the surrounding atmosphere of the sampler and the gas-absorbing surface. Other factors influencing the diffusion and the uptake rate of the passive sampler are ambient air temperature and the thickness of the diffusion layer in the vicinity of the absorbing surface, respectively. The diffusion layer thickness changes with variable air flow velocities along the sampler, i.e., increasing air flow enhances the transfer of the gaseous compound to the absorber and leads to a higher uptake of the target compound for a given target gas concentration at a fixed sampling time. Since in practical applications of passive sampling, the airflow can generally not be controlled and in outdoor measurements depends on the wind speed, the relation between gas-phase concentrations and the uptake of the analyte gas may not be well defined. What is actually measured in such cases is the deposition-velocity-sensitive accumulated deposition flux, also termed the immission rate (see below), of the target gas. The dimension of this value is deposited mass per time and area rather than mass per volume for gas-phase concentrations. This issue was, for instance, discussed in the context of studies about the deposition of corrosive gases on metallic surfaces [27]. Also, a German standard method exists determining the immission rates of gases using a continuously recirculating absorber solution [28]. Obviously, the accumulated diffusion flux (or immission rate) is connected to the gas-phase concentration, but it is not the same.

A breakthrough in passive sampling can be seen in a paper published by Palmes et al. in 1976 [29]. They introduced a simple tubular sampling device with a chemical trap for the analyte gas at the bottom of the tube. The tube-type configuration was designed in a way that the diffusion path length becomes defined by the length of the tube. If the length is sufficiently large (precisely the aspect, i.e., the ratio between length and inner diameter of the tube is higher than about 5–8), wind shortening effects caused by eddies entering the tubes become insignificant [2,7]. With the additional feature of the sampler that the absorbing surface acts as a perfect sink for the analyte gas, i.e., the gas concentration at the surface gets and remains zero during the sampling period, the gas-phase concentration can be derived from Fick's first law of diffusion [1,2,7]. Hence, gas-phase calibration of the samplers for analyte quantification is not required. At about the same time as propagating the tubular design of passive samplers, badge-type configurations with a short distance between the open side and the analyte collecting surface were introduced [30–34]. Early applications of the badge-type samplers were personal exposure measurements in workplace atmospheres [1,33,34], but later on, and until today, badge-type samplers are also used for ambient air measurements [35–37]. The short diffusion path length of badge-type samplers causes increased sampling rates, but wind shortening due to turbulent transfer of the analyte gas caused by wind incursion into the open-end of the sampler is more likely to occur. To counteract wind effects, the open side of the badge-type passive samplers is generally either covered by a gas-permeable polymeric membrane (so that gas transport is no longer governed by diffusion rather than permeation) or porous plugs are inserted into the short diffusion path to act as a turbulence barrier [2,4,5,38]. In many of the published

papers and also in commercially available passive samplers the one or the other means has been implemented. These measures, however, require evaluating the uptake rate of such samplers experimentally for proper quantification of target gas concentrations using calibration with standard gases and/or comparison with established reference methods.

Colorimetric gas detectors and gas sensors have been widely applied in many fields, such as industrial hygiene and process control, as well as ambient and indoor air quality monitoring (e.g., [39–42]). The basic concept behind this is to measure the color change of a sensing zone that occurs in the presence of a target analyte gas. The gas transport to the sensing zone can occur actively by using a pump. This has been realized in the majority of applications since it provides higher sensitivity and better precision compared to passive sampling [43]. However, exposure of colorimetric gas sensors to the sample gas without forced convection in a passive sampling mode has all the advantages mentioned above, and a number of publications make use of these features (e.g., Chapter 4 of [43] and [44,45]).

The configurations of colorimetric gas sensors are diverse. They encompass flat sheets of different base materials (often paper or polymer membranes), tubes filled with the sensing material, and badges with inner support onto which the color-forming reagents are immobilized [39,40,42,43]. In all these sensors, the gaseous compounds of interest are absorbed or adsorbed at the collection zone, where a selective chemical reaction takes place, leading to a colorization; in cases where the reagent is colored, it gets decolorized. The color change and/or the change of color intensity are used as analytical information. As outlined above, for passive samplers, the diffusional transport to the sensing zone can or cannot occur through a defined air gap by permeation through a suitable membrane. Hence, the distinction between colorimetric gas sensors delivering concentration information and results about the accumulated deposition flux or immission rate can—or should—be made. It appears worth mentioning already at this stage that in publications of passive sampling methodologies and colorimetric gas sensors, this fact has—to the best knowledge of the authors—never been considered (see below).

One pre-requisite for quantifying gas-phase concentrations using passive sampling is the quantification of the collected amount of the target gas or a derivative. To this end, a variety of procedures and detection methods have been used. A distinction can be made between procedures where (i) determination occurs within the passive sampler, (ii) the collected derivative is extracted after the sampling step by a suitable solvent followed by determination in the liquid phase, and (iii) thermodesorption of volatile compounds and their transfer to, e.g., gas chromatography [4,5,7]. In the two former procedures the collected compounds/derivatives are often quantified by spectrophotometric methods. In cases where the absorber is or contains a color-forming reagent, direct in situ measurement can be performed, and conceptionally, this is the preferential option [46]. However, more frequently, the colored compounds are extracted, followed by spectrophotometric detection in the extract. If the collected derivative is colorless the color reaction can either be initiated within the sampler body before or in a reaction vessel after liquid extraction.

Regarding color detection in general, and also for evaluation of passive samplers a large variety of methodological and instrumental configurations exist. Qualitative information can be obtained by visual inspection with the naked eye. Extension of visual detection using color cards/charts as a reference provides instrument-free semi-quantitative measurements. Quite a number of passive samplers (often also termed passive dosimeters, see above) in tube and badge configurations for gas analysis—many of them being commercially available—use this approach (e.g., [47–53]). For true quantitative detection, spectrophotometric and, more rarely, reflectometric detection is applied either in situ or after (liquid) extraction [9,10,46,52,53].

An attractive alternative to common optical detection in general is the application of digital color measurements (a common term for that is digital image colorimetry, DIC). DIC refers to a colorimetric analysis method based on digitizing images collected by some image acquisition tools such as mobile phones, digital cameras, webcams, scanners, and dedicated color reading devices [54]. Several techniques (i.e., absorbance and reflection

measurements, fluorometric detection) of digital color measurements (often using different terminology even when the underlying principles are the same or very similar) for analytical chemical applications have been introduced and presented in numerous reviews [55–59]. The application of digital color imaging for analytical chemical purposes has undergone explosive development in recent years. The main analytical applications are related to the medical, biochemical, nutritional, and environmental fields [55,56,59]. Compared with other means of digital imaging (i.e., scanners, digital cameras, webcams, color readers with RGB sensors), smartphones are widely used as image acquisition tools in DIC due to their ubiquitous presence, convenient use, remarkable and continuing improvement of the camera functions, and the ability to adopt mobile applications for data evaluation and transmission [58–60]. Not least because of these attractive features, we have limited our review to applications of smartphone-based color evaluation of passive samplers for gas analysis.

Color parameters generally evaluated are RGB, HSV, Greyscale color systems, and Euclidean distance. Some smartphone applications (e.g., Photometrix, Color Grab) provide retrieving the respective color values, but more common and flexible is to export the images and to employ programs like ImageJ, Photoshop, GIMP, etc. for evaluation.

The use of smartphones for color detection in the evaluation of passive sampling of gases has been presented in quite a number of papers. However, the term passive sampling is only rarely used in the title (and often also not throughout the entire text) of relevant publications. A closer inspection of all papers where gas sampling was performed passively by molecular diffusion followed by smartphone-based color detection and signal evaluation revealed that there are severe differences regarding the design of the sampling devices and the way how color detection and evaluation have been performed. A conceptual difference also exists between samplers that are suitable for gas concentration measurements and those providing information about the accumulated deposition flux of the gaseous compounds of interest. As mentioned already above, the former type requires a defined and sufficiently long diffusion path or adopts turbulence barriers in front of the absorbing layer to avoid wind-shortening effects. Within this review, we will name passive samplers of this configuration type-1; the others with uncontrolled (air velocity dependent) diffusion path lengths are accordingly termed type-2 passive samplers. The design, analytical specifications, and applications of these two types will be separately presented in the body of this paper. Figure 1 provides a general scheme of the individual steps of the methodology, starting with (i) passive sampling via (ii) color formation (or decolorization) reactions, (iii) smartphone photographing, (iv) color retrieval, and evaluation, to (v) calculation of results. Differences referring to the type and geometry of the passive samplers, illumination conditions of photographing, means of image processing, and the kind of quantitative information eventually obtained are also implemented in Figure 1 in a somewhat generalized manner.

The aim of this review is to present for the first time a compilation of the still few publications devoted to smartphone-based color evaluation of passive samplers for gases. Though the underlying concept is rather simple, the instrumental and procedural details of the published work differ in many respects. Also, various gases have been determined using different chemical reactions for color formation or discoloration. Therefore, we attempt to present a comprehensive description of the work performed so far with a critical discussion of features and limitations. Since we are convinced that the methodology inherently provides a number of very attractive features for gas analysis, the review may also act as a stimulus for other scientists to contribute to further developments, refinements, and areas of application.

Before coming to the title subject of this review, some remarks on terminology, differentiation, and delimitations about passive sampling of gases and various procedures involved in color detection of the collected compounds are presented.

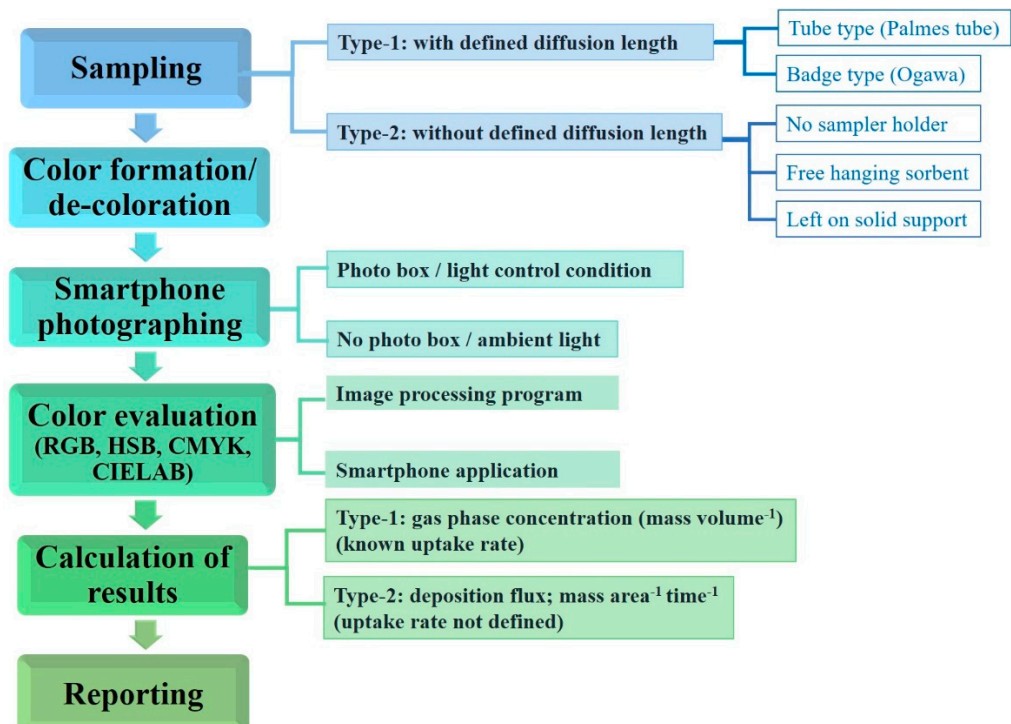

**Figure 1.** General scheme of the individual steps of the methodology and differences referring to type and geometry of passive samplers, illumination conditions of photographing, means of image processing, and the kind of quantitative information obtained.

## 2. Remarks on Terminology of Passive Sampling Devices and Conceptional Distinction of Analytical Evaluation Procedures

This chapter appears necessary to us because the terminology relating to passive sampling is anything but harmonized or standardized, and this may create confusion for readers. For example, the term passive sampling is often used in publications, but severely different sampler and gas sensor configurations and evaluation procedures are presented or, vice versa, very similar (almost identical) configurations and procedures are referred to using different terms. Additional confusion may arise from the fact that the term passive dosimeters [61,62] is also used instead of passive samplers (see also above). This, however, appears quite appropriate since, with almost all passive sampling devices (including gas sensors), time-weighted average information is achieved about the uptake (dose) of the sampler or the sensing zone. Only in rare cases does the response of passive sampling devices depict real-time (momentary) changes in the sample gas uptake. Another item that, in our opinion, deserves consideration is the way of evaluation of all passive sampling devices regarding the required chemical reactions for color formation or discoloration involved and the optical detection techniques applied.

The adjective "passive" is used in connection with passive sampling to distinguish it from active sampling. The latter includes (in the context of gas analysis) the supply of air to the collection medium or a measuring system by suction with a pump or provision using a blower. "Passive", on the other hand, means that no intentional convection is generated to control the gas transport rather than that the gaseous analytes reach the absorber mainly due to molecular diffusion. Convection caused by natural air movement always exists (due to thermal gradients or wind) and will hence overlap with the diffusion process [2]. Because of the predominantly diffusion-controlled transport to the sorbent surface, the term diffuse sampling is often used synonymously with passive sampling. It is worth mentioning that there are numerous publications in which passive/diffusive collection of gases obviously takes place, but the respective methods are not referred to as such.

The term sampling also requires an explanation or definition in the context of passive sampling. In the narrower sense, sampling only involves the collection of a sample and tells nothing about the substances of interest. In passive sampling, however, it is not the matrix gas (air, atmosphere, etc.) that is collected, but usually, a specific substance or several substances simultaneously is/are trapped and collected more or less selectively. The term collection suggests a kind of enrichment (accumulation) of the target component or a derivative, which is, in fact, what happens in most—but not all—cases of passive sampling. Sometimes, conversion of the target gas in contact with the absorber occurs, leading directly to detectable (colored) compounds. In such cases, the term gas sensor—considering the IUPAC definition of sensors: "A chemical sensor is a device that transforms chemical information, ranging from the concentration of a specific sample component to total composition analysis, into an analytically useful signal" [63]—appears appropriate. However, it can be stated again that the terminology is often inconsistent, and the term "sensor" is also used for a variety of devices that involve (active and passive) sampling followed by, for instance, subsequent temporal and locally separated determination of the analyte using diverse analytical methods. Often, sensors are also defined only when the immediate and reversible response to a target compound occurs [64].

A distinction can also be made with regard to the analytical procedure of color evaluation and quantification. The three main routes are as follows: (i) A color change of the sorptive medium occurs in contact with the target gas. To this end, the absorbing surface itself is or contains a reagent that (preferentially) selectively forms the colored compound or is decolorized when a colored absorbing reagent is used. This allows for direct reading with the naked eye or evaluation by color or color intensity comparison with a scaled color card. This is, for instance, a common practice in direct reading gas dosimeters for personal monitoring [57,58], where time-weighted exposure information is obtained. Evaluation of gas dosimeter tubes is typically made by length-of-stain readings [65,66]. For badge-type geometries, a pre-printed color scale located beside the sensing zone is commonly used for semi-quantitative concentration readings. Many colorimetric gas sensors can also be included in this category. Often, they are directly exposed to the sample air and transport to the sensing surface is by molecular diffusion. In all mentioned instances, qualitative and semi-quantitative information is accessible using color cards as the reference, but quantitative determination generally requires calibration with standard gases. (ii) A color-forming reagent is added to the sampling device after the sampling period, and detection takes place (in situ) within the passive collector. As before, qualitative and semi-quantitative information can be achieved by visual inspection and color comparison with a reference color card. For quantitative information, photometric or reflectometric detection is applied. The former, however, is in many instances handicapped by geometrical restrictions of positioning the light source and detector. In order to derive the gas-phase concentration, either calibration with standard gases is required, or in cases where the uptake can be calculated by Fick's first law of diffusion the determination of the amount of derivative collected is sufficient. This is readily obtained by liquid-phase calibration of the photometric or reflectometric procedure. (iii) After sampling, the target component or a derivative is extracted, and the extract is removed from the sampler, followed by subsequent derivatization and color detection. This is by far the most common procedure of color evaluation of passive samplers. One advantage can be seen in the ease of using photometric detection (eventually, the colored solutions are transferred into a cuvette). It is also possible to apply different methods for the analysis of the extract, and repeated measurements on the same extract can be made for control purposes. Drawbacks are the many procedural steps involved and the loss of sensitivity due to the dilution of the collected compound in the extract.

## 3. Presentation of Selected Publications

In the following two Sections, an outline of publications about passive sampling devices (including gas sensors) according to type-1 and type-2 samplers (see above) is presented in which smartphone evaluation has been applied for color detection. We at-

tempted to cover this subject matter comprehensively. To this end, a search in existing databases (i.e., Google Scholar, SciFinderSearch, Web of Science, Scopus) was conducted using combinations of the search terms "smartphone", "digital imaging", "passive sampling", "dosimeters", "color sensors", "air analysis" and "gas analysis". We have intentionally limited the current presentation solely to smartphone-based color evaluation, although other digital devices (i.e., digital cameras, scanners, webcams, dedicated RGB sensors, etc.) have been employed for evaluation of passive samplers, dosimeters, and color sensors and often provide similar, sometimes even improved features [46,67–71]. The limitation to smartphone-based color evaluation was made since the number of existing publications would otherwise go beyond the scope of an exhaustive review. Another reason was the fact that smartphones are presently always at-hand, are convenient to use, can be easily connected to the internet, and provide options for the implementation of apps for data treatment. Potentially, the evaluation of smartphone images can be readily performed by non-specialists in the frame of experiments in student courses and/or citizen science projects.

Figure 2 illustrates schematically some of the individual steps of the entire procedure presented in Figure 1. Since the instrumental configurations and procedures applied in the respective publications that are presented below differ in many respects the figure cannot represent all variants. It only serves as an example that encompasses the basic steps from passive gas sampling to color evaluation.

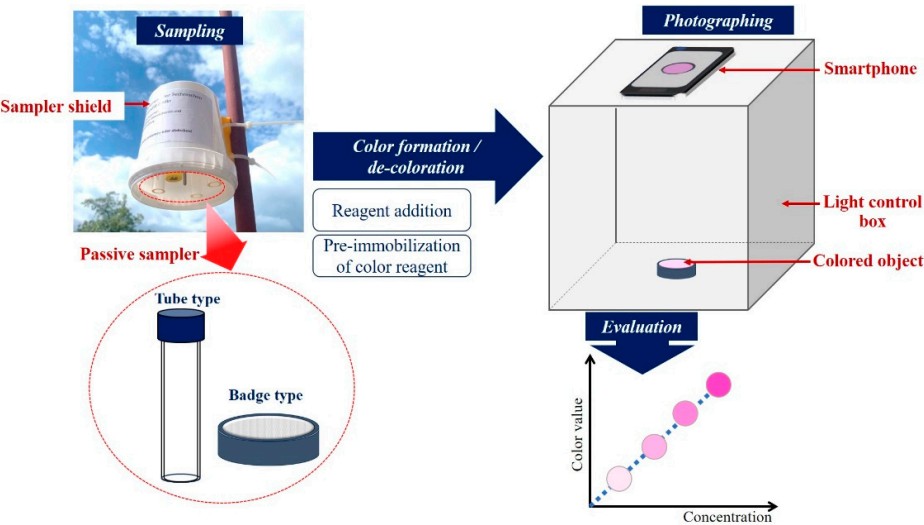

**Figure 2.** Illustration of the individual steps of the methodology of smartphone-based color evaluation of passive gas sampling (see text for further explanations).

The selected papers of the categories type-1 and type-2 are sorted by date of publication. An overview of the in-total 14 publications is given in Table 1. The respective configurations of the devices applied in the selected publication and the way they are exposed to the sample air are described in the following. Emphasis will also be laid on the analytical procedures regarding the color reaction as well as the smartphone detection and color evaluation procedures. If available, the performance of the samplers regarding analytical specifications (working range, detection limits, selectivity) is outlined. In papers where real sample analysis has been reported by the authors, this will be presented together with means of validation and results achieved.

**Table 1.** Compilation of publications referring to smartphone-based evaluation of passive sampler for gases/vapors. A distinction is made between Type-1 and Type-2 samplers (see text for further information). Publications are sorted by the date of publication.

| Year | Analyte Gas | Sorbent Fabrication | Sampler Geometry | Reagent | Detection Condition | Photographing Condition | Evaluation Software | Color System | Working Range | LOD | Application | Ref. |
|---|---|---|---|---|---|---|---|---|---|---|---|---|
| | | | | | **Type-1** | | | | | | | |
| 2020 | Ozone | pre-immobilized coloring reagent on sorbent | commercial passive samplers (Owaga badge) | Indigo | direct detection of fading of blue color | photo box | Corel DRAW X5 and Matlab software (version R2015a) | RGB | 11–109 µg m$^{-3}$ (exposure time not given) | 3.3 µg m$^{-3}$ | Suburban environment | [72] |
| 2021 | Nitrogen dioxide | immobilized trapping reagent into sorbent pad | lab-made passive sampler (tube type) | Griess-Saltzman reagent | adding Griess-Saltzman gel | photo box | ImageJ (version 1.52e) | RGB | not given | 32 µg m$^{-3}$ (24-h exposure) | No real sample analysis | [73] |
| 2023 | Nitrogen dioxide | immobilized trapping reagent into sorbent pad | lab-made passive sampler (Palmes tube) | Griess-Saltzman reagent | adding pre-mix reagents (Griess-Saltzman) | without a photo box (ambient light) | ImageJ (version 1.53a) | RGB | 10–120 µg m$^{-3}$ (14 days exposure) | 5 µg m$^{-3}$ (14 days exposure) | Urban environment | [74] |
| | | | | | **Type-2** | | | | | | | |
| 2016 | Hydrogen sulphide | pre-immobilized coloring reagent on sorbent | no sampler holder | N, N-Dimethyl-p-phenylenediamine, and Fe (III) | direct detection of methylene blue product | photo box | GIMP software (version 2.8) | CMYK | 5–50 ppm (30 min exposure) | 0.12 ppm (30 min exposure) | Sewage treatment plant | [75] |
| 2016 | Formaldehyde | pre-immobilized coloring reagent agar sorbent | no sampler holder | 4-Amino-3-hydrazino-5-mercapto-1,2,4-triazole, ZnO, KIO$_4$ | direct detection of color change | without a photo box (ambient light) | Adobe photoshop (not given) | RGB | 20–85 µg m$^{-3}$ (24-h exposure) | 11 µg m$^{-3}$ (24-h exposure) | Indoor air (formaldehyde emission flux) | [76] |
| 2017 | Mercury vapor | pre-impregnated Corning porous Vycor glass reagent on sorbent | no sampler holder | Nanogold | direct detection of color change | not given | not given | RGB | uptake 0.06–0.6 µg | not given | Personal sampling of miners | [77] |
| 2018 | Mercury vapor | pre-immobilized cuprous iodide/polystyrene composite on sorbent | no sampler holder | Cuprous iodide/polystyrene composite | direct detection of color change | photo box | ImageJ (version 1.49 hr) | RGB | 61–270 µg m$^{-3}$ (30 min exposure) | 16 µg m$^{-3}$ (30 min exposure) | No application reported | [78] |
| 2018 | VOC | pre-immobilized reagent on sorbent | cap of vial | Polydiacetylenes | direct detection of color change | not given | Adobe photoshop/Android Studio app (not given). | RGB | not given | not given | Identification of VOC | [79] |

**Table 1.** *Cont.*

| Year | Analyte Gas | Sorbent Fabrication | Sampler Geometry | Reagent | Detection Condition | Photographing Condition | Evaluation Software | Color System | Working Range | LOD | Application | Ref. |
|---|---|---|---|---|---|---|---|---|---|---|---|---|
| 2021 | Hydrogen sulphide | pre-immobilized reagent on sorbent | encapsulated between two glass plates | Arene-derivative dye | direct detection of color change | without a photo box (ambient light) | Adobe photoshop (version CS6) | CIELAB, RGB, HSB and CMYK | 0–1.5 ppm (15 min exposure) | not given | No application reported | [80] |
| 2021 | Hydrogen sulphide | pre-immobilized coloring reagent on the surface of the glass substrate | no sampler holder | Indium oxide nanostructure | direct detection of color change | photo box | Colorimetric Detector application (not given) | Optical darkness ratio | not given | 10 ppm (30 sec exposure) | No application reported | [81] |
| 2021 | Ammonia | pre-immobilized reagent on paper sorbent | not given | Methyl orange | direct detection of color change | photo box | ColorAssist app (version 2.1, FTLapps) | HIS | 6.0–54.0 ppb (3 min exposure) | 2 ppb (exposure time not given) | Chicken farm | [82] |
| 2021 | Ammonia, formaldehyde, hydrogen sulfide | pre-immobilized (screen-printing) reagent on polymer-coated paper | no sampler holder | Bromocresol green, fluorescent dye (primary amine), cupper azo complex | direct detection of color change | photo box | Time-lapse app (not given) | RGB | not given | not given | No application reported | [83] |
| 2023 | Hydrogen sulphide | pre-immobilized reagent on agarose hydrogel | The cap of a centrifuge tube | Copper (II)-azo complex | direct detection of color change | not given | ColorAssist app (not given) | RGB | 1–50 ppb (10 min exposure) | 43.34 ppb (10 min exposure) | Exhaled breath | [84] |
| 2023 | Ozone | pre-immobilized reagent on polydimethyl siloxane sheet | no sampler holder | o-Dianisidine | direct detection of color change | photo box | ImageJ (not given) | RGB | 0–200 ppb (8 h exposure) | 1.79 ppb (8 h exposure) | Printing store, rubber molding press factory, residential house | [85] |

*3.1. Passive Samplers of Type-1*

A thorough review of all papers found in the literature search (see above) revealed that only three publications (one of them recently submitted) fulfill the above-mentioned criteria for this categorization. A brief outline of the content of these papers is given in the following.

(1) A low-cost and portable method for the monitoring of daily tropospheric ozone levels has been reported by Cerrato-Alvarez et al. [72] through the combination of passive sampling and digital analysis of images taken with the camera of a smartphone. Commercially available Ogawa passive samplers were employed in all measurements. The analytical signal used was the degree of decolorization of the blue color of indigotrisulfonate (ITS) deposited on cellulose filters upon the reaction with ozone. Photos were taken in a homemade photobox with controlled internal luminosity by a strip containing 14 bright LEDs. To prevent the influence of possible inhomogeneous degree of decolorization, the image of the whole pad was saved for digital analysis in PNG format employing CorelDraw and MATLAB software. RGB parameters were obtained from images taken with the camera. The red color channel was selected for quantification since it is the complementary color of the reflected blue color of the ITS solution. Evaluation of grey-scale intensity, effective absorbance, and Euclidean distance were tested with marginal differences regarding the specifications achieved. The ozone concentration in ambient air was derived from Fick's first law of diffusion. The mass of ozone reacted during the sampling time is calculated stoichiometrically from the equivalent amount of ITS consumed during the sampling period, obtained from digital image analysis of the sampling pads. ITS amount deposited on the pads and the exposure time were adjusted to fit different expected ozone levels in ambient air. Under optimized conditions, the linear range obtained was between 11 and 109 $\mu$g m$^{-3}$ with a detection limit of 3.3 $\mu$g m$^{-3}$. Precision and accuracy of RSD = 6.8% and relative errors ranging from $-14.0$ to 5.7%, respectively, are stated. The practical application of the method was tested by measuring 24-hour average levels of ozone in a suburban environment over a period of three months. Correlation against a spectrophotometric reference method was r = 0.77. According to the authors, the potential of the method as an auxiliary tool to other methods for ozone determination could be demonstrated, providing a rapid and decentralized measurement of ozone levels with adequate reliability. Another important advantage of the presented method mentioned is that the analysis can be performed by anyone without the need to be a specialized analyst. Therefore, the developed method may help to increase community awareness and commitment to air quality issues.

(2) In a paper by Souza et al. [73], a passive sampling method is described for the determination of nitrogen dioxide through the formation of a colored dye, followed by digital image analysis of the resulting color intensity. The passive samplers were purpose-made from conical Falcon tubes used in chemical laboratories cut to the desired size. The gas collection was made using a mixture of triethanolamine (TEA), ethylene glycol, and acetone as the sorbent immobilized on Whatman cellulose filters of 2.6 cm diameter. For determination of the nitrite formed during adsorption of NO$_2$ in TEA, a gel containing the Griess-Ilosvay reagent was distributed across the collecting surface with a plastic ruler, whereafter the filters were placed in a photobox equipped with high-brightness LEDs for illumination, and pictures were taken with a smartphone. RGB colors were evaluated by ImageJ software. Uptake rates of the samplers were not measured and could not be calculated on the basis of Fick's law due to the design of the samplers using a polymeric membrane as a turbulence barrier. In order to obtain quantitative NO$_2$ data, parallel measurements have been conducted initially with on-line chemiluminescence monitors at the same site. This permitted a correlation of the intensity of the color generated in the passive sampling method with ambient NO$_2$ concentrations. Results of digital image analysis and spectrophotometric evaluation after extraction as a reference statistically agreed at a 95% confidence level. The authors state that the advantages of the technique include low cost, the ready availability of components, ease of use, and sensitivity. The achievable detection limit stated is 32 $\mu$g m$^{-3}$ NO$_2$ for a 24-h sampling period. The authors

conclude that the method could be applied for both outside and indoor environments, in particular for low budget laboratories. Real sample analysis has, however, not been made.

(3) A smartphone-based color evaluation procedure of tube-type passive samplers has been exemplarily investigated for the determination of $NO_2$ in a paper by Shi et al. [74]. The purpose-made samplers were similar to Palmes' tubes with TEA immobilized on cellulose filters used for collection of $NO_2$. Thorough optimization of various experimental parameters affecting the color evaluation has been made. One of the specific aims of the authors was to avoid the use of a photo-box (commonly used in most papers referring to smartphone-based color detection) for photographing the colored objects since this practically detracts from at-site analysis. Photographic parameters investigated were conditions of illumination and distance between the smartphone and the colored objects both found to be partially interrelated. In order to minimize solution handling and transfer, the color-forming Griess-Ilosvay reagent was added directly to the adsorber pads and smartphone photos were taken (in situ) of the colored liquids contained in the cap of the passive sampler tube. In this context, additional parameters investigated were the composition and volume of the reagent and the color of the caps. Smartphone photos were saved in JPEG format, and RGB values of the colored solutions were retrieved using the freeware ImageJ. The green color channel (the complementary color to the pink reaction product of the nitrite assay) was used for evaluation. Calibration was performed using liquid standards prepared and processed in the same way as the sample solutions. By taking photos of samples and standards simultaneously, the influence of variable illumination conditions could be eliminated.

As a result of the optimization, high sensitivity of nitrite determination (nitrite being the product of sampling $NO_2$ with TEA) with a working range of 30–600 ng, a limit of detection (LOD) of 12 ng, and good precision (<8% RSD over the entire concentration range) was attained. For a sampling duration of 14 days and using the tube-type samplers of the present work, the working range and LOD for atmospheric $NO_2$ are 5–200 $\mu g\,m^{-3}$ and 1.8 $\mu g\,m^{-3}$, respectively. The experience gained during the principal investigation and optimization was exemplarily put into practice in two small measurement campaigns determining ambient $NO_2$ in the city of Berlin (Germany). Reasonably good agreement was achieved with the data presented by the governmental air pollution control network in Berlin.

### 3.2. Passive Samplers of Type-2

The thorough review of all papers dealing with smartphone-based evaluation of passive samplers found in the literature search (see above) revealed that the presentations of 11 publications can be categorized as type-2 devices. The description of the design of the passive sampling devices (including color sensors) and the way of installation during the sampling step evidenced that the uptake of analyte gas is indeed governed by molecular diffusion, but no attempts were made (or not described) to obtain a controlled length of the diffusion path. As mentioned above, this fact detracts from concentration measurements because of the uncontrolled length of the diffusion path due to wind effects. Despite this, the application of Fick's law of diffusion has sometimes been falsely used for the calculation of the sampler's analyte uptake, and analyte gas concentrations in $\mu g\,m^{-3}$ are given rather than the appropriate value of the immission rates in $\mu g\,m^{-2}\,h^{-1}$ (see above). In the following, the contents of the 11 publications on type-2 passive sampling devices are presented in a condensed form.

(4) The development of a low-cost colorimetric sensor for the determination of $H_2S$ using smartphone-based evaluation is reported by Pla-Tolós et al. [75]. A mixture of N,N-Dimethyl-p-phenylenediamine, $FeCl_3$, and glycerol has been immobilized on Whatman filter paper circles of 3 cm diameter. In the presence of the $H_2S$, methylene blue is formed, which can be detected visually. The reaction product obtained was found to be highly stable in this support and is free of blank signals. For quantitative estimation, smartphone imaging (among other optical methods) has been applied. In passive sampling experiments, $H_2S$

gas was prepared in a 2 L bottle. The concentration was estimated based on the generation of $H_2S$ from a known amount of $Na_2S$ acidified with HCl. The sensor circles are suspended on a thread in the bottle and exposed to the gas-phase for 30 min. After exposure, the sensor circles were washed with water to remove the excess reagent. A smartphone camera was employed to take pictures of the sensors. Conditions of photographing, i.e., use of a photobox or ambient light, illumination, and geometrical arrangement, are not given. The color picker tool of GIMP was used to evaluate the color intensity of the pictures. The CMYK (cyan, magenta, yellow, black) color-coordinate system was applied to convert the images into numerical color values. A calibration graph was constructed as a plot of the value of the system readout vs. $H_2S$ concentration. A working range of 5–50 ppm $H_2S$ and a LOD of 1.12 ppm has been achieved. The proposed procedure was applied to the determination of air samples in the vicinity of a sewage treatment plant in Comunidad Valenciana (Spain).

(5) In a paper by Sekine et al. [76], the development of a novel colorimetric formaldehyde detector applied in a passive sampling configuration is reported using the built-in camera of a mobile phone for evaluation. The colorimetric detector employs a solid phase colorimetric reagent made from 4-amino-3-hydrazino-5-mercapto-1,2,4-triazole, ZnO, $KIO_4$, and agar. A color change of this reagent occurs from white to purple by exposure to HCHO gas. The colorimetric performance of the detector was first assessed by exposure of the sensor to HCHO vapor from a droplet of HCHO aqueous solution in a closed Petri dish. Unfortunately, the support material for the color-forming reagent, the size of the sensing zone, and the geometrical arrangement of the sensor relative to the formaldehyde source are not given. HCHO vapor generated moved towards the colorimetric reagent within a headspace by molecular diffusion, and the gas molecules then reacted with the reagent. A digital image of the detector was taken using a smartphone camera positioned 15 cm above the desk level at ambient light (no photobox used). It was found that the calibration of the measured color intensity with a color standard reduced the variation of the results. The influence of different mobile phones for imaging and changing conditions of photographing (in particular kinds of illumination) were also investigated. To quantify the color change of the detector, the color was converted to a color value of the green channel in a RGB color model on a personal computer. Adobe Photoshop was used for calculation. For calibration purposes, the detector was placed in a small test chamber, and the response of the detector to known concentrations of HCHO in air was investigated with a constant gas generation system. The working range of the sensor for 24-h exposure is 20–85 $\mu g\ m^{-3}$ HCHO with a limit of quantification of 11 $\mu g\ m^{-3}$. The authors state, that this meets the requirements to detect the indoor air quality guideline level of HCHO set by the World Health Organization. The developed detector was also applied to classify HCHO-emitting building materials, i.e., plywood, whose emission flux is greater than 120 $\mu g\ m^{-2}\ h^{-1}$.

(6) A passive sampler with smartphone evaluation for monitoring of gaseous elemental mercury in artisanal gold mining has been presented by de Barros Santos et al. [77]. The passive sampling devices were made from rods of Corning porous Vycor glass (PVG). The rods were cut (using a diamond disc) to obtain small PVG discs of 0.1 cm in thickness and 0.6 cm in diameter. A very thin gold layer was deposited onto the surface by impregnation of PVG discs with a $HAuCl_4$ solution, followed by in situ reduction of $AuCl_4$ to elemental Au, using sodium borohydride as a reduction agent. The presence of nanogold was readily detected visually by the red color appearance of the PVG/Au disc. Preliminary experiments were conducted by placing PVG discs in the upper part of a glass vessel (resembling a test chamber), and a drop of liquid mercury was given to the bottom of the vessel. The red color tones changed with increasing exposure times to the Hg vapor atmosphere, and this was used as the analytical signal. The experimental set-up of taking photos with the smartphone (e.g., geometrical arrangement) and illumination conditions are unfortunately not detailed in the paper. The variation of the RGB (Red, Green, Blue) color patterns of the PVG/Au discs was plotted in histograms, and results were compared to discs (used as control) that were not exposure to mercury vapor. RGB histograms showed that the red

color channel is more sensitive to the amount of mercury than the green and blue channels. To obtain quantitative data on the mercury retention, both Au and Hg in each sampler were quantified in the lab by using ICP-MS und Direct Mercury Analysis (DMA) techniques, respectively. In this way, it was shown that the PVG/Au sampler can detect the uptake of Hg in the range between ~0.06 to 0.6 μg. Calculation of mercury vapor concentrations in air was not intended since the aim of the work was the estimation of personal exposure. It is worth mentioning that in the presented configuration, the results obtained refer only to the uptake of a person's skin or clothing rather than quantitative information about the amount of inhaled mercury vapor. The performance of the PVG/Au sampler was evaluated in a simple field application in an artisanal and small-scale gold mining environment in Burkina Faso. To this end, the PVG/Au samplers were placed on the front lapels of the shirts of miners who were present at the Au–Hg amalgam burn.

(7) Salcedo et al. [78] developed a device for colorimetric determination of mercury vapor using smartphone camera-based imaging. It consists of a sensing zone based on a cuprous iodide/polystyrene composite exhibiting a reddish color in the presence of elemental Hg vapor. The sensing layer was prepared by mixing CuI powder with a polymeric binder solution in tetrahydrofuran. The resulting emulsion was applied onto a Whatman chromatography paper through a roll-coating technique using a glass rod. The colorimetric sensing paper was then cut into small pieces and set on a glass slide for easier handling during the experiments. The colorimetric paper sensors were placed inside glass vials, which were then capped with a silicone septum. Different volumes of standard $Hg^0$-saturated air were injected into the vials using a syringe. This allowed the exposure of the colorimetric paper sensors to varying concentrations of Hg vapor. Upon exposure to $Hg^0$, the color of the colorimetric sensing paper sheets changed from white to light orange. For digital image acquisition, the colored sheets were placed in a light-tight box to avoid the influence of varying ambient light on sensor illumination. Stable lighting was provided by a fluorescent tube located at the upper portion of the lightbox. The photos taken after exposure to $Hg^0$ were analyzed in the RGB color space using the open-source image processing program ImageJ. Percent change in the red, green, and blue values of the sensor before and after exposure to $Hg^0$ was calculated. The blue-based response was eventually used for calibration. The linear working range of the CuI/polystyrene composite sensors is reported to range between 61 and 270 $\mu g\,m^{-3}$ $Hg^0$. The calculated limit of detection was 16 $\mu g\,m^{-3}$ $Hg^0$. Application to real sample analyses is not reported.

(8) Park et al. [79] have developed a smartphone-based colorimetric paper sensor for the qualitative detection of volatile organic compounds using an array of polydiacetylenes (PADs) as color-forming compounds. The array was formed of four different PDAs on conventional paper using inkjet printing of the corresponding diacetylene monomers, followed by photopolymerization. For testing vapochromism, a small volume of each solvent was poured into a plain glass vial, and the vial was sealed by a cap and incubated for 30 min at ambient temperature. Upon opening the cap, a punched part of the PDA sensor array was inserted immediately into the vial, allowing exposure to the saturated solvent vapor. After closing the cap, the time evolution of the color change (typically blue-to-red) in the PDA array was recorded over 30 min using a smartphone camera, and the images were analyzed using photoshop software or an Android Studio app. Unfortunately, the conditions of photographing (e.g., camera settings, illumination) and the selection of the region of interest of the colored spots have not been detailed. Exposure of the PDA array to an unknown solvent promotes color changes, which are imaged. A database of color changes (i.e., the red channel of the RGB color space and hue values) was then constructed on the basis of different vapochromic responses of the 4 PDAs to 11 organic solvents. To this end, a new "combinatorial" strategy (no details are given in the manuscript) was devised, taking into account the different response behavior of the PDAs to various solvents. Subsequently, this enabled the identification of a particular volatile organic solvent with high accuracy.

(9) A paper-based color sensor for sensitive and selective detection of gaseous $H_2S$ has been presented by Vargas et al. [80]. The sensing zone contains an arene-derivative dye embedded into a porous cellulose matrix. This paper strip containing the sensing probe was then encapsulated between two glass plates attached to each other. In this way, the chromatography paper was covered by the glass on both sides, producing a channel into which the gas could easily and homogeneously penetrate by molecular diffusion. In the stage of optimization of the procedure, exposure of the color sensor to the gas was performed in a cuvette. Gaseous $H_2S$ was prepared in situ by the addition of HCl to $Na_2S$ aqueous solutions within the cuvette. The sensor was fixed to one of the walls of a quartz cuvette, keeping it out of contact with the liquid that was placed in the lower part. After exposure, the sensors were removed, and pictures of the sensors were taken in raw image format using the camera of a regular smartphone. Constant and uniform artificial light (without a photobox) was used for all pictures, with a white chromatography paper strip placed beside the sensor as a reference for the comparative analysis of the pictures and to control for possible light fluctuations. Processing of the photos was performed using Adobe Photoshop without applying any color or exposure corrections. The color information for each of the sensor exposures was extracted using the major color spaces (i.e., CIELAB, RGB, HSB, and CMYK). For an exposure time of 15 min, saturation of the color is reached above ~1.5 ppm $H_2S$, but a linear response was found between 0 and 1.5 ppm. The authors emphasize that with the present sensor, it is possible to perform direct calibration at low $H_2S$ concentrations through the color extraction of digital pictures taken with a common smartphone, broadening the potential range of use of the disposable sensor. Unfortunately, an application to real sample analysis and validation of the developed method is not reported.

(10) A colorimetric gas sensor for determination of $H_2S$ was developed by Devi et al. [81]. The sensor is based on indium oxide ($In_2O_3$) nanostructures, which have been prepared by a modified sol–gel method. A sensing nanostructured film is obtained by spray coating method onto 1 cm $\times$ 1 cm glass plates. Exposure of $H_2S$ to the sensing surface causes a color change due to the sulfurization of the top-layer of the $In_2O_3$ nanostructured film and the formation of an $In_2S_3$ layer. In the experimental setup, a mobile phone was fixed in a dark wooden cabinet and photos of the colorimetric detector were captured by an Android mobile phone at constant illumination. To quantify the colorimetric sensing of $H_2S$ gas detection, the optical darkness ratio (ODR), of the sensor has been followed by a dedicated smartphone-based application. At room temperature the lower limit of detection of $H_2S$ gas by the $In_2O_3$ nanostructured film was 10 ppm for an exposure time of 30 s. The selectivity of the sensor over NO, $NO_2$, CO, $N_2$, Ar, and $NH_3$ was high. Application to real samples has not been done.

(11) Khachornsakkul et al. [82] reported a paper-based colorimetric device for the on-site screening of ammonia gas. The detection principle is based upon a color change from red to yellow of methyl orange immobilized on a paper substrate. After exposure to ammonia gas for 3 min, photos were taken with a smartphone of the colored substrate in a photobox. The color signal of the device has been measured through the hue channel of an HSL system via the application of a smartphone. The hue values and degree from the HSL system on the paper sensor were obtained by using the software ColorAssist installed in an iPhone. An advantage of the hue evaluation was emphasized in that it is not relative to the intensity and brightness of the occurred color; therefore, this channel can reduce errors from these influences. Calibration was made by generating $NH_3$ gas from the evaporation of aqueous $NH_4OH$ solution and placing the sensor in the headspace above this solution. The preparation method of $NH_3$ gas standards was validated using an electrochemical gas sensing instrument. The linear relationship between $NH_3$ concentration and the hue signal of the sensor was from 6.0 to 54.0 ppbv with a 2 ppbv limit of detection. The applicability of this device was demonstrated in the determination of $NH_3$ in a laboratory and at a chicken farm. Since the color change of the pH-indicator is fully reversible, the recorded color only represents the momentary uptake of ammonia gas and not, as most other passive

samplers and color sensors, a time-weighted average. This fact has not been mentioned by the authors and clearly limits the sensors' applicability.

(12) A method has been developed by Engel et al. [83] to monitor the exposure to different gases (viz. ammonia ($NH_3$), hydrogen sulfide ($H_2S$) and formaldehyde (HCHO)) in ambient air. The method is based on a visible color change of colorimetric gas sensors, which can be evaluated by the naked eye, a stationary color reader, or the camera of a smartphone. The sensors consist of a disposable paper tag or plastic card and gas-sensitive materials, which have been deposited by a screen-printing process. The integration of the gas-sensitive layers into a machine-readable pattern of a QR-like code incorporating color reference spots provides illumination, camera-independent calibration, and quantitative detection. For $NH_3$ and HCHO detection, commercially available pH-sensitive color dyes have been employed. $H_2S$ is detected by an immobilized copper(II) azo dye complex. The color change of the gas-sensitive layer due to the reaction with the target gas was characterized by the evaluation of RGB values taken with an in situ readout station (unfortunately, it is not explained in the paper what this is and how it works) using the camera of an iPhone and a commonly available time-lapse app, taking consequent consistent images every five seconds. The determination of the color values of individual pixels was implemented with the help of a Python script. The readout station contained in a transparent gas measurement chamber was built in an opaque plastic box to achieve constant illumination using several white LEDs. Information on how the sensors are fixed within the gas measurement chamber is lacking.

In the cases of detection of $NH_3$ and $H_2S$, the indicator reactions are reversible. Therefore, only the momentary response to varying gas concentrations is obtained (and not the commonly achieved time-weighted values of passive sampling devices). The color dye 4-amino-3-penten-2-one selected as a colorimetric sensing material for the detection of formaldehyde, forms a fluorescent dye, which turns from colorless to neon yellow when it comes in contact with the target gas. The reaction involved is also reversible, but due to the very slow (within days) return to the colorless form, the color sensor has—according to the author's opinion—potential for the preparation of disposable dosimeters. None of the presented sensors has been employed for real sample analysis.

(13) A colorimetric sensor for $H_2S$ detection using smartphone-based color evaluation has been presented by Wang et al. [84]. The sensing zone was constructed by incorporating copper(II) pyridine diazinonaphthol (Cu-PAN) complex into agarose hydrogel. The reaction of $H_2S$ with the reagent leads to a color change from purple to yellow, which has been used as analytical information. A small portion of the gel was pipetted into the cap of a 10 mL centrifuge tube. For calibration, $H_2S$ gas was quantitatively generated by the reaction between $Na_2S$ and HCl within the tube. Hence, diffusive sampling occurs from the headspace above the liquid phase. After a sampling time of several minutes, the cap was removed, and photos were taken of the colored gel using a smartphone camera. Conditions of photographing, i.e., ambient light or photobox, geometrical arrangement, etc., are not presented. The color change was read out by recording RGB values and data collected with the help of the Color Assist app on a smartphone. Euclidean distance was applied for quantification of the color intensity. The response of the color sensor showed good correlation with the logarithm of $H_2S$ concentration in a wide range from below 1 ppm to about 50 ppm for 10 min sampling time. A limit of detection of 43.34 ppb is stated. Possible interference by various gases was tested, resulting in the high selectivity of the developed sensor. Long-term stability was also high. The feasibility of the Cu-PAN hydrogel sensor for the measurement of $H_2S$ levels in human exhaled breath was demonstrated.

(14) Passive sampling of ozone with colorimetric detection using o-Dianisidine as the sorbent has been reported by Choi et al. [85]. The reagent was immobilized in polydimethylsiloxane (PDMS) sheets, which were cut to the desired size and served as a collector for ozone. o-Dianisidine, a colorless compound, undergoes a visible color change to yellow upon contact with $O_3$. Optimization and calibration were performed by smartphone photographing of the sheets placed in a photobox illuminated with a white LED. The captured

images were processed using ImageJ. The entire area of the sheets was selected using a polygon selection tool, and the average red, green, blue (RGB), and greyscale intensity were recorded using an RGB measure plugin. Exposure studies of the samplers were conducted in a test chamber with known $O_3$ concentration in the range 0–200 ppb for variable durations up to 8 h. The passive sampling sheets were calibrated by measuring the absorption of o-Dianisidine after liquid extraction of the collecting sheets in exposure experiments under the same conditions. Colorimetric changes were analyzed by capturing the images obtained from smartphone photographing, and the effective absorbance of the blue scale was shown to provide the best fit for changing $O_3$ concentrations. Limits of detection and quantification of 1.79 ppb and 5.27 ppb $O_3$, respectively, are stated. The selectivity of the passive sampler was examined by exposure to several other gases potentially present in indoor environments, but no interferences were found. Based on the optimization experiments, badge-type passive samplers were constructed and fixed at the lapel for personal exposure studies. The samplers were employed in several field tests conducted in a printing store, a rubber molding press factory, and a residential house. The results obtained in the printing store evidenced a significant disparity between $O_3$ concentrations within the room and personal $O_3$ exposures. The use of a smartphone app with warning information at high $O_3$ exposure is mentioned in the paper, but no details are presented. The authors conclude that the developed passive sampling methods can increase awareness of health-threatening $O_3$ exposure among workers and occupants.

### 4. Conclusions, Critical Remarks, and Outlook

Passive sampling of gases and vapors is a well-established method for the determination of time-weighted average concentrations. It is widely employed in surveillance of ambient and indoor quality and is also used for personal monitoring of exposure to problematic air pollutants in industrial hygiene. In passive sampling, the gaseous analytes are trapped on suitable sorbents, whereby often a colored derivative is formed, the intensity of which is used for quantitative detection. In other instances, the derivatives are extracted, and a color reaction is initiated in a separate vessel prior to color intensity measurements for quantification. The most common method for color detection is spectrophotometry, although reflectometric detection has also been used. Colorimetric gas sensors often operate in a passive sampling mode (no pumps are used for the provision of air to the sensing surface), so they (despite the different and inconsistent terminology) are obviously a kind of passive sampling device with color detection for qualitative or quantitative gas determination. In part of the present review, conceptional similarities and distinctions are discussed.

With the advent of color imaging using digital cameras, webcams, scanners, dedicated color readers, and smartphones, these instruments are increasingly used for the evaluation and quantification of color and color intensities (not only) in the context of chemical analysis. Smartphone-based evaluation is particularly attractive due to the ready availability of smartphones, their convenient use, their small size, portability, the ever-increasing quality of the camera, and dedicated apps that can be implemented for data presentation and transmission. Not least because of these features (yet also to keep the number of publications manageable), we have limited our review to smartphone-based evaluation applications for passive samplers.

The thorough search of publications in which smartphone-based evaluation of passive sampling devices (including colorimetric sensors for gases) is presented has finally led to only 14 relevant papers. Determination of various gases and vapors are described, i.e., $NO_2$, $O_3$, $NH_3$, $H_2S$, Hg vapor, formaldehyde, and toxic organic volatiles. The inspection of these publications revealed considerable differences with respect to the design of the sampling devices, the way they are exposed to the sample air, the color evaluation procedures of the photos taken with a smartphone, and the calculation of results from color change or color intensity variations. One very important aspect raised in our review is the distinction between passive sampling devices that provide concentration information of the target

gases (due to a defined length of the diffusion path), and those where the uptake of the respective gas is indeed connected to gas concentration but what is really measured is the time-weighted deposition flux (also termed immission rate, see above) since the diffusion path varies with variable air movement in the vicinity of the samplers. This will happen with changing wind speed in ambient air measurements or movement of humans in personal monitoring applications. It is interesting (even surprising) that in none of the publications, this fact has been discussed, and in most of the publications, gas concentrations in, e.g., $\mu g\ m^{-3}$ are given rather than the appropriate value for target gas deposition flux in mass per area and time. In our review we have made a clear separation between these two situations by categorizing the passive sampling devices presented in the selected publications in type-1 and type-2 and have outlined the contents of the respective papers in two separate chapters.

Regarding the design of the passive sampling devices, both tube-type and badge-type configurations have been used in various publications. Only in one paper, a commercially available passive sampler has been employed, whereas in all other papers, purpose-made samplers have been constructed. Unfortunately, details of the dimensions and way of incorporation of the sorbent into the samplers are sometimes missing. Different procedures of passive sampling and smartphone-based color evaluation for various target gases have been reported in various publications. A deficiency of many papers can be seen in missing and difficult-to-understand description (or sometimes inadequate) calibration procedures. And even when devices have been calibrated this has been performed often in laboratory experiments, but how these data have been used for quantification in real sampling situations remained unclear.

The sorbent materials and chemical reactions responsible for color development (or in two cases of decolorization) are well described in all papers. What is not always adequately presented in the respective publications is the experimental set-up for exposure measurements and subsequent conditions of taking smartphone images of the colored or decolorized zones. Photos taken with the smartphone are generally exported to a computer and different imaging processing software (often freeware ImageJ) has been used to select appropriate regions of interest. Only in one case the inherent capability of the smartphone was employed. The RGB color space is most often applied for quantification of the color response of the passive sampling devices; in some papers, CIELAB, HSB, and CMYK have been used instead or supplementary. However, not all papers present the complete relevant information in a way that makes it clear how the records have been made. Calibration procedures employed in the various procedures differ considerably. They include the use of standard gases, calculations based on color changes achieved in liquid-phase measurements of corresponding compounds to the respective gaseous analytes, comparison with other determination methods for the same gas applied in parallel measurements, and comparison with prefabricated color charts. In some papers, the calibration procedures are very well described; in others, the description is only rudimental and hard to comprehend. The application of the developed passive sampling devices to real sample analysis is presented in 9 of the 14 publications. The other five papers are regarded to be either only a proof of concept or—at least in the opinion of the authors of this review—could not convincingly demonstrate the feasibility of practicable application of gas analysis.

Considering the current features of digital color imaging in general and smartphone-based photographing of colored objects in particular, it comes as no surprise that this has already attracted considerable attention for analytical chemical applications. The relatively few published works related to the evaluation of passive samplers for gases and colorimetric gas sensors operating in a passive (diffusion-controlled) sampling mode should not be regarded as a sign of limited advantageous features of this approach rather than the yet missing recognition of this possibility. In our opinion, the selected papers presented here clearly evidence significant potential for measurement of various trace gases and vapors exceeding the ones that have been dealt with so far.

A certainly very valuable application area of smartphone-based evaluation of passive samplers and colorimetric gas sensors is at-site in situ color detection and quantification. Considering the low cost and simple construction of passive sampling devices (many of them can be self-made with low efforts from readily available materials) and the ubiquitous presence of smartphones all over the world, it enables citizens to measure air pollutants in their immediate environment and will probably enhance the awareness of possible risks to health and environment [18,86].

For a couple of years, we have applied smartphone-based color evaluation of passive sampling for the determination of $NO_2$ in student courses of the curriculum of the department of environmental chemistry and air research. The feedback from the students was very positive, and some of the students later elaborated, modified, and improved configurations in the frame of Bachelor and Master Theses [87,88]. The current work of our groups in Berlin and Chiang Mai focuses on the miniaturization of passive sampling devices involving 3D-printing technology and further simplification of the color formation procedure with ready-to-use spray reagents. An attempt has also been made to develop a dedicated app for RGB color detection and data recording, documentation, and transmission of results via the internet.

Additional future directions of research regarding the concept of smartphone-based color evaluation of passive samplers should focus on (i) (further) simplification of the analytical procedures regarding the preparation of the reactive layer of the samplers and the post-sampling process of color formation, (ii) suitable (i.e., convenient and reliable) calibration procedures implementing, e.g., pre-printed color scales, and (iii) application to real-life measurements and validation of results. Finally, the distribution of information about the potentialities of the methodology away from the sole publication in scientific journals is another item that should be increasingly considered by researchers to motivate the public dealing with analytical chemical problem solving and the role of healthy air for breathing.

**Author Contributions:** Conceptualization, W.F.; writing—original draft preparation, K.K. and W.F.; writing—review and editing, K.K., K.G., A.H. and W.F.; visualization, K.K. All authors have read and agreed to the published version of the manuscript.

**Funding:** This research was funded by the Alexander von Humboldt Foundation, Georg Forster Research Fellowship of grant number [Ref 3.5—THA—1227646—GF-P], and a renewed research stay grant number [Ref 3.5—THA/1009402].

**Institutional Review Board Statement:** Not applicable.

**Informed Consent Statement:** Not applicable.

**Data Availability Statement:** No new data were created or analyzed in this study. Data sharing is not applicable to this article.

**Acknowledgments:** K. Kiwfo appreciates the support from a Georg Forster Research Fellowship of the Alexander von Humboldt Foundation (Ref 3.5—THA—1227646—GF-P) for funding research conducted at TU Berlin (Germany). K. Grudpan thanks the Alexander von Humboldt Foundation for funding a renewed research stay at TU Berlin (Ref 3.5—THA/1009402).

**Conflicts of Interest:** The authors declare no conflicts of interest.

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
