# Peer review of "Smartphone-Based Color Evaluation of Passive Samplers for Gases: A Review"

_atmosphere, doi:10.3390/atmos15040451_

Round 1

Reviewer 1 Report

Comments and Suggestions for Authors

Comments

This review shows the different works published in the literature in the last years on the application of smartphone-based color evaluation of passive sampling devices for gases.

These types of papers are very interesting to know the most relevant advances on the use of smartphone as analytical devices for color detection and quantification and their application to different target gases. This review can be accepted for publication with major changes.

- Include in the introduction a paragraph with the color parameters used in the literature as an analytical signal. Also, include smartphone applications used to analyze images, e.g. photometrix.

-Include some Figures showing the schemes of methodologies used in some of the references cited.

- The number of references used in this review are too few. I consider that it is necessary to increase the references in “Passive sampler of type-1”.

-As it is mentioned in the conclusion, publications have been limited to smartphone-based evaluation applications for passive samplers. Introduce a paragraphs about smartphone as analytical tool in the introduction, and reference reviews such as,

https://doi.org/10.1016/j.trac.2023.117284

https://doi.org/10.1016/j.aca.2015.10.009

https://doi.org/10.1016/j.trac.2019.06.019

Author Response

This review shows the different works published in the literature in the last years on the application of smartphone-based color evaluation of passive sampling devices for gases.

These types of papers are very interesting to know the most relevant advances on the use of smartphone as analytical devices for color detection and quantification and their application to different target gases. This review can be accepted for publication with major changes.

- Include in the introduction a paragraph with the color parameters used in the literature as an analytical signal. Also, include smartphone applications used to analyze images, e.g. photometrix.

Reply to comment 1: A few sentences have been added to the introduction mentioning the typical color parameters used as analytical signal and also examples of programs for evaluation/analyzing the color images.

-Include some Figures showing the schemes of methodologies used in some of the references cited.

 Reply to comment 2:

Each referred reference is already discussed in the text. So, schemes of the method and/or figures would give repetition and are hence not necessary.

- The number of references used in this review are too few. I consider that it is necessary to increase the references in “Passive sampler of type-1”.

 Reply to comment 3: The reasons of distinction between type-1 and type- 2 passive samplers were explained in Chapter 2 “Remarks on terminology of passive sampling devices and conceptional distinction of analytical evaluation procedures”. Considering this, we have found (based on a thorough review of the available literature) that only the three publications cited meet the respective criteria.

-As it is mentioned in the conclusion, publications have been limited to smartphone-based evaluation applications for passive samplers. Introduce a paragraphs about smartphone as analytical tool in the introduction, and reference reviews such as,

https://doi.org/10.1016/j.trac.2023.117284

https://doi.org/10.1016/j.aca.2015.10.009

https://doi.org/10.1016/j.trac.2019.06.019

Reply to comment 4: One reference mentioned by the referee was already cited (Ref. No. 56). The two others have been added as suggested by the reviewer (now Refs. 60 and 61).

Reviewer 2 Report

Comments and Suggestions for Authors

The use of smartphone-based detection technology is currently a hot topic and is considered highly significant. Smartphones possess high-performance capabilities for quantifying color. Moreover, they are characterized by their compact size, affordability, and widespread availability, allowing for various analyses through applications accessible to anyone. In the field of analytical chemistry, research utilizing smartphones is abundant. However, in the area of atmospheric environment and gas measurement, there are still relatively few studies utilizing this technology, and contributions such as research papers are welcomed.

The authors of this review provide a thorough review of the current state of research, which is valuable.

However, authors should check the comments below.

1. Table 1and 2: There are line breaks in the middle of gas name. Do not break the line.

2. The number of line 311: The same word ("software") appears twice. Delate one word.

3. Reference 73: Was following reference paper accepted? If not, this paper should not be referenced.

Shi C.; He X.; Kiwfo, K.; Held, A.; Frenzel, W., Optimization of smartphone-based evaluation of tube-type passive samplers using atmospheric nitrogen dioxide determination as an example, J. Environ. Sci. Health A, 2023, submitted

4. The number of line 604- 606.

In follow sentence, is the comma (,) after the camera a period (.)?

After a sampling time of several minutes the cap was removed and photos were taken from the colored gel using a smartphone camera, Conditions of photographing, i.e. ambient light or photobox, geometrical arrangement etc., are not presented.

5. The number of line 619:

   "o-dianisidine" should be written as "o-Dianisidine"  

Author Response

1. Table 1and 2: There are line breaks in the middle of gas name. Do not break the line.

 Reply to comment 1: Table has been modified according to reviewer’s suggestion.

2. The number of line 311: The same word ("software") appears twice. Delate one word.

 Reply to comment 2: Doubling of the word “software” has been removed.

3. Reference 73: Was following reference paper accepted? If not, this paper should not be referenced.

Shi C.; He X.; Kiwfo, K.; Held, A.; Frenzel, W., Optimization of smartphone-based evaluation of tube-type passive samplers using atmospheric nitrogen dioxide determination as an example, J. Environ. Sci. Health A, 2023, submitted

 Reply to comment 3: Ref. 75 has been re-submitted to another journal. As we could see in previous publications in “Atmosphere” references to a submitted article has obviously been accepted.

4. The number of line 604- 606.

In follow sentence, is the comma (,) after the camera a period (.)?

After a sampling time of several minutes the cap was removed and photos were taken from the colored gel using a smartphone camera, Conditions of photographing, i.e. ambient light or photobox, geometrical arrangement etc., are not presented.

 Reply to comment 4: The comma in line 606 has been changed for a dot (it was a printing error).

5. The number of line 619:

   "o-dianisidine" should be written as "o-Dianisidine"  

 Reply to comment 5: We have corrected for “o-Dianisidine”.

Reviewer 3 Report

Comments and Suggestions for Authors

This review (atmosphere-2914139) focuses on smartphone-based color evaluation of passive samplers for gases. The organization is basically complete and reasonable, but the readability is poor and there are many problems to be solved before possible publication. My specific comments are as follows:

1.         Title: (1) The first letter of all words needs to be capitalized. (2) Colorimetric gas detectors and gas sensors are the core, is smartphones the key core? Smartphones are just a technology. Using smartphones as a limitation resulted in limited materials for related references.

2.         The readability of this review is poor, and it is recommended to combine relevant data and figures.

3.         Suggest discussing the working principle of colorimetric gas detectors and gas sensors.

4.         Introduction: The logic of the manuscript is unclear. In terms of the current development status of sensors, the electronic gas sensors are still the mainstream. Suggest a comprehensive and brief discussion (such as resistance type, Chemosensors 2024, 12(3), 43; Optical type, Sens. Actuators A Phys. 367 (2024) 115052), followed by the introduction of colorimetric gas sensors and detection technologies (such as low energy consumption, no need for batteries, and convenience).

5.         Suggest elaborating and discussing different types of colorimetric gas sensors.

6.         “Conclusion” section needs to be strengthened. Readers would like to see more constructive strategies and prospects of colorimetric gas sensors and detection technologies.

7.         Please carefully check the format/style of the target journal. For example, the reference format. The numbers in the chemical formula require subscripts, including references.

8.         English writing of the manuscript needs to be polished.

Comments on the Quality of English Language

English writing of the manuscript needs to be polished.

Author Response

This review (atmosphere-2914139) focuses on smartphone-based color evaluation of passive samplers for gases. The organization is basically complete and reasonable, but the readability is poor and there are many problems to be solved before possible publication. My specific comments are as follows:

Reply: It appears to us, that the reviewer has partially misunderstood the intention of the present review. The main issue is passive sampling of gases and evaluation of colors formed by reactions between gases and a sorbent/reagent using smartphone imaging. The relation/overlap between the term passive samplers and colorimetric gas sensors has been outlined in Chapter 2 on “Remarks on terminology of passive sampling devices and conceptional distinction of analytical evaluation procedures”. From this chapter it should also become clear why we have made a distinction between type-1 and type-2 sampler configurations.

1. Title: (1) The first letter of all words needs to be capitalized. (2) Colorimetric gas detectors and gas sensors are the core, is smartphones the key core? Smartphones are just a technology. Using smartphones as a limitation resulted in limited materials for related references.

Reply to comment 1:

(1) Amendment has been made accordingly.

(2) The core of the MS is - as the title states – smartphone-based color evaluation of passive samplers. The limitation to smartphone as a color detector was done because of the otherwise to extensive number of publications. Also, smartphone color evaluation (as compared with other digital imaging possibilities) is very attractive for reasons mentioned in the MS (see lines 169-173).

2. The readability of this review is poor, and it is recommended to combine relevant data and figures.

Reply to comment 2: It is unclear to us what the reviewer has meant specifically. Since no figures are present, combing relevant data and figures cannot be made.

3. Suggest discussing the working principle of colorimetric gas detectors and gas sensors.

Reply to comment 3: Gas sensors in general and colorimetric gas sensors are not the core of the MS. In fact, as mentioned above, only colorimetric gas sensing in combination with passive sampling and smartphone color evaluation have been considered. A few references regarding colorimetric gas sensors are already part of the MS (see Ref. 39-42).

4. Introduction: The logic of the manuscript is unclear. In terms of the current development status of sensors, the electronic gas sensors are still the mainstream. Suggest a comprehensive and brief discussion (such as resistance type, Chemosensors 2024, 12(3), 43; Optical type, Sens. Actuators A Phys. 367 (2024) 115052), followed by the introduction of colorimetric gas sensors and detection technologies (such as low energy consumption, no need for batteries, and convenience).

Reply to comment 4: We cannot well follow the suggestion by the reviewer. Certainly, he is right in mentioning that many other sensor principles exist but since the focus of the MS is on passive sampling with smartphone-based color evaluation a general treatment of sensors and their features appears too far-reaching and hence inappropriate.

5. Suggest elaborating and discussing different types of colorimetric gas sensors.

Reply to comment 5: The manuscript focuses on passive samplers for gases utilizing smartphone-based evaluation. Again, we want to state that a presentation of colorimetric gas sensors (and discussion about configurations and respective features) is out of the scope of the MS.

6. “Conclusion…” section needs to be strengthened. Readers would like to see more constructive strategies and prospects of colorimetric gas sensors and detection technologies.

Reply to comment 6: Once again; strategies and prospect of colorimetric gas sensors are not within the scope of our MS. Many publications exist (some being cited in our MS, see above) which deal with colorimetric gas sensors and optical evaluation methods. Our focus was on the rather recent emergence of smartphone-based color evaluation and was limited (as mentioned above) only to passive gas sampling.

7. Please carefully check the format/style of the target journal. For example, the reference format. The numbers in the chemical formula require subscripts, including references.

Reply to comment 7: We have carefully checked all formalities and made appropriate corrections.

8. English writing of the manuscript needs to be polished.

Reply to comment 8: The English writing has been checked again by all authors of the MS and a bilingual (German/English) colleague and is considered by us to be well done.

Round 2

Reviewer 1 Report

Comments and Suggestions for Authors

Title: Smartphone-based color evaluation of passive samplers for gases: A review

Comments

This review shows the different works published in the literature in the last years on the application of smartphone-based color evaluation of passive sampling devices for gases. These types of papers are very interesting to know the most relevant advances on the use of smartphone as analytical devices for color detection and quantification and their application to different target gases. This review can be accepted for publication without change.

Author Response

Reply to comments of Reviewer 1

According to reviewer 1 no additional changes are required.

Reviewer 3 Report

Comments and Suggestions for Authors

The following issues have not been substantially resolved. Additionally, it is recommended to submit a response in the form of a document attachment.

1.      As a review paper, the attractiveness of this manuscript is insufficient. Suggest the authors to add figures from references. The saying goes, one picture is worth a thousand words.

2.      Lack of papers, only listing the literature. What is the contribution of this manuscript? Peer researchers can easily search for these literature through Google Scholar.

3.      The detection principle, challenges, and future development directions have not been clearly explained.

Comments on the Quality of English Language

Minor editing of English language required.

Author Response

The reply to reviewer 3 is attached .
